The Bacteria Genome Pipeline (BAGEP): an automated, scalable workflow for bacteria genomes with Snakemake

Olawoye Idowu B. 1 2
Frost Simon D.W. 3 4
Happi Christian T. happic@run.edu.ng 1 2
1 Department of Biological Sciences, Faculty of Natural Sciences, Redeemer’s University , Ede , Osun State , Nigeria
2 African Centre of Excellence for Genomics of Infectious Diseases (ACEGID), Redeemer’s University , Ede , Osun State , Nigeria
3 Microsoft Research , Redmond , WA , USA
4 London School of Hygiene & Tropical Medicine, University of London , London , United Kingdom
Aramayo Rodolfo
Electronic publication date: 2020 Oct 27
Publication date: 2020
Volume: 8
Electronic Location ID: e10121
Received 2020 Apr 29; Accepted 2020 Sep 16
Copyright: ©2020 Olawoye et al.
Copyright year: 2020
Copyright holder: Olawoye et al.
License: This is an open access article distributed under the terms of the Creative Commons Attribution License, which permits unrestricted use, distribution, reproduction and adaptation in any medium and for any purpose provided that it is properly attributed. For attribution, the original author(s), title, publication source (PeerJ) and either DOI or URL of the article must be cited.
License URL: https://creativecommons.org/licenses/by/4.0/

Keywords: Bacteria genomics, Bioinformatics, Pipeline, Workflow

Funding: African Centre of Excellence for Genomics of Infectious Diseases (ACEGID) The World Bank ACE019 The National Institute of Health 5U01HG007480-03 U54HG007480 ELMA FluLab This work was supported by African Centre of Excellence for Genomics of Infectious Diseases (ACEGID), the World Bank (ACE019) and The National Institute of Health grants 5U01HG007480-03 and U54HG007480 to Christian T. Happi. This research was also supported by ELMA, and the FluLab as part of the TED-Audacious project funding support to Christian T. Happi. The funders had no role in study design, data collection and analysis, decision to publish, or preparation of the manuscript.

==============================
Next generation sequencing technologies are becoming more accessible and affordable over the years, with entire genome sequences of several pathogens being deciphered in few hours. However, there is the need to analyze multiple genomes within a short time, in order to provide critical information about a pathogen of interest such as drug resistance, mutations and genetic relationship of isolates in an outbreak setting. Many pipelines that currently do this are stand-alone workflows and require huge computational requirements to analyze multiple genomes. We present an automated and scalable pipeline called BAGEP for monomorphic bacteria that performs quality control on FASTQ paired end files, scan reads for contaminants using a taxonomic classifier, maps reads to a reference genome of choice for variant detection, detects antimicrobial resistant (AMR) genes, constructs a phylogenetic tree from core genome alignments and provide interactive short nucleotide polymorphism (SNP) visualization across core genomes in the data set. The objective of our research was to create an easy-to-use pipeline from existing bioinformatics tools that can be deployed on a personal computer. The pipeline was built on the Snakemake framework and utilizes existing tools for each processing step: fastp for quality trimming, snippy for variant calling, Centrifuge for taxonomic classification, Abricate for AMR gene detection, snippy-core for generating whole and core genome alignments, IQ-TREE for phylogenetic tree construction and vcfR for an interactive heatmap visualization which shows SNPs at specific locations across the genomes. BAGEP was successfully tested and validated with Mycobacterium tuberculosis (n = 20) and Salmonella enterica serovar Typhi (n = 20) genomes which are about 4.4 million and 4.8 million base pairs, respectively. Running these test data on a 8 GB RAM, 2.5 GHz quad core laptop took 122 and 61 minutes on respective data sets to complete the analysis. BAGEP is a fast, calls accurate SNPs and an easy to run pipeline that can be executed on a mid-range laptop; it is freely available on: https://github.com/idolawoye/BAGEP.

Introduction

Over the years, as next generation sequencing has rapidly become popular, molecular biology has taken a giant leap in the way genomes of various organisms are deciphered. Genomics have grown expeditiously with high throughput sequencing technologies and paving the way for novel biological analytical approaches (Schuster, 2008)

Genome sequencing of pathogens such as bacteria and viruses generate FASTQ files which contains quality scores of the nucleotide bases in raw format. These files usually need various technical processing steps such as adapter removal, sequence quality filtering and quality control. This is crucial for other downstream analysis, as seen in a work where choosing the appropriate FASTQ pre-processor improved downstream analysis significantly in comparison to other software (Chen et al., 2018).

In addition, reconstructing a sequenced genome is usually achieved through de novo assembly (genome assembly using overlapping regions in the genome) or through a reference sequence guided approach (mapping sequence reads to a reference genome) (Korbel et al., 2007; Zhang et al., 2011). There are numerous tools that can perform these tasks at various efficiency and for different lengths of sequence reads. A few genome assemblers such as SPAdes, Burrows-Wheeler Alignment tool (BWA), minimap2 and Bowtie are widely used in bioinformatics (Bankevich et al., 2012; Langmead & Salzberg, 2012; Li, 2018; Li & Durbin, 2009).

Variants in microbes occur as single nucleotide polymorphisms (SNPs), insertions and deletions (indels) and/or translocations. Mapping the reads to a reference genome or comparing the assembled genome to a reference genome are the ways to detect variants (Olson et al., 2015). SNP detection is important for comparative genomics of bacterial isolates as they have been used to characterized different bacterial species sharing the same genus (Ledwaba et al., 2019). SNP analysis is also an integral part to understand evolution of bacterial genomes, detecting the cause or source of an outbreak, phylogeography and genome-wide association studies (GWAS) (Farhat et al., 2019; O’Neill et al., 2019; Stimson et al., 2019).

At the moment, there is no generally accepted standard operating procedure for evaluating and calling SNPs and/or indels which has led to a wide variety of tools and methods for variant detection. Furthermore, best practices for variant identification in microbial genomes have been proposed in literature and it has been adopted to a large extent (Olson et al., 2015; Van der Auwera et al., 2013).

Downstream analysis of numerous bacterial genomes using whole genome sequencing (WGS) pipelines require high performance computing servers or a cloud-based support system. In addition, while a number of WGS pipeline exist such as UVP (Ezewudo et al., 2018) and MTBseq (Kohl et al., 2018) for Mycobacterium tuberculosis, these pipelines require huge computing infrastructure. Moreover, the pipelines usually have many dependencies to be installed especially if the analysis requires multiple tasks to be performed such as phylogenetic tree construction, drug resistance prediction, clustering, and so on. The huge amount of bioinformatics tools available makes it a daunting task in picking the suitable software for a certain analysis. In low-to-middle income countries who are becoming users of WGS of pathogens, it is also important to provide ready-to-use, quick and easily deployable pipelines for WGS data analysis. These pipelines should also be a stand-alone workflow on a local server or even a personal computer. For example, UVP and MTBseq needs at least 100 GB and 25 GB of RAM respectively to run locally which is capitally intensive for many researchers in low-to-middle income countries (LMICs) who need to analyze WGS data.

To address the challenges identified above, we embarked on the development of BAGEP (Bacteria Genome Pipeline, available online at https://github.com/idolawoye/BAGEP#). It uses existing pipelines and bioinformatics tools and an advanced workflow management system called Snakemake (Koster & Rahmann, 2012). This pipeline combines many tools used in downstream analysis of paired-end FASTQ raw reads such as quality control, read mapping, variant calling, core genome and/or full genome alignments, SNP visualization and phylogeny into a single, fast, easy-to-use workflow.

Materials & Methods

Sampling

To display the versatility of BAGEP, we applied it to a set of multi-drug resistant (MDR) M. tuberculosis genomes (n = 20) from Southwest Nigeria (Senghore et al., 2017) and S. enterica serovar Typhi genomes (n = 20) from the United Kingdom (Ashton et al., 2017). These publicly available paired-end FASTQ reads were extracted from National Centre for Biotechnology Information (NCBI) Sequence Read Archive (SRA). Genomic DNA of isolates were extracted from liquid cultures and sequenced on Illumina MiSeq and HiSeq platforms. These raw reads are deposited under BioProjects PRJEB15857 and PRJNA248792.

Implementation

Installation of BAGEP requires the pipeline to be downloaded on to a personal computer and installed by creating a conda environment to set up all dependencies. Centrifuge database should be downloaded separately in order to detect contaminants in the samples. Complete installation steps are stated in the github README file: https://github.com/idolawoye/BAGEP/blob/master/README.md.

The analysis of BAGEP is segmented into set of ‘rules’ that connects the output file of an analysis into the input of the next task in the general workflow (Fig. 1). The dependencies are fastp for quality control of raw reads, Centrifuge for taxonomic classification, Snippy for variant detection, snippy-core for core and whole genome alignments, Abricate for AMR detection, IQ-TREE for phylogenetics, Krona for taxonomic visualization, vcfR and heatmaply for SNP visualization.

Figure 1 A schematic workflow of BAGEP.

Fastp is used for quality control on paired-end FASTQ reads. The processed reads are mapped against a reference genome provided by the user. Centrifuge is used to check the reads for contamination and generate a taxonomic visualized report with Krona. Variants are called using Freebayes and annotated with SnpEff (aided by Snippy). The resulting variant call format (VCF) files and genomes from each sample are collated with Snippy-core to produce a VCF file containing all samples, core and whole genome multiple sequence alignments. A maximum-likelihood phylogenetic tree is constructed with IQTREE and a HTML file containing an interactive SNP visualization with the VCF file. Finally, Abricate generates an AMR report from whole genome multiple sequence alignments.

These tools can be installed in a containerized manner using a bioconda channel (Dale et al., 2018) which can be activated and deactivated easily, whereas the SNP visualization is provided by R libraries included in the dependencies. The input files for BAGEP are paired-end FASTQ files and a reference genome in FASTA or GenBank format, only the latter is provided in the single line command to run, whilst the former should be saved in a designated local directory.

BAGEP also allows full customization of the pipeline, such that users can modify the parameters used in running their samples. For example, the pipeline does not mask repeat regions when aligning the core genomes by default, but this can be done by adding the option to the snakefile rule handling that process. It is possible to modify every step of the workflow to suit the samples being processed, even the model for phylogenetics.

Upon execution of the pipeline, Snakemake organizes the correct combination of tasks in order to generate the desired output and also leverages on the wildcard feature to analyze multiple genomes that have similar format allowed in the rule (Koster & Rahmann, 2012). Preliminary results such as quality trimmed FASTQ files are saved in a new directory, while primary results like detected variants, core genome alignments, phylogeny and SNP visualization are saved in a different directory. The pipeline runs on Linux/Unix, MacOS terminal and Windows Subsystem for Linux. However, no prior skill in programming is required to run and interpret the results as only one stage of input is needed and thereafter the pipeline runs automatically. Another significant advantage of BAGEP is that it can re-execute steps that have not been ran if errors occur and resume from where it stopped rather than restarting the entire process which another valuable feature of the underlying framework.

Results

FASTQ pre-processing

BAGEP performs several quality control tasks on raw FASTQ reads using fastp (Chen et al., 2018). Fastp capitalizes on speed, efficiency and an all-in-one pre-processor. It carries out sliding window quality filtering, adapter trimming, base correction, polyG and polyX tail trimming for Illumina NextSeq and NovaSeq series, UMI pre-processing to reduce background noise and improve sensitivity when detecting ultra-low frequency mutations in deep sequencing, duplication analysis, quality control and reporting (Fig. 2).

Figure 2 Pre-processing reports from read 1 of one of the M. tuberculosis genomes.

(A) Read quality before filtering. (B) Read quality after filtering. (C) Base contents before filtering. (D) Base contents after filtering.

Fastp was chosen as the suitable tool for this task as it is seen to outperform other quality control FASTQ software and it also allows multi-threading to improve its performance when dealing with numerous samples (Chen et al., 2018).

Taxonomic classification

This step is crucial in the pipeline as contaminants from the environment such as host cells or other bacteria can negatively impact the accuracy of variants detected. We utilized Centrifuge for this step due to its lower memory requirements compared to other similar tools like Kraken (Kim et al., 2016). To visualize the results, Krona was employed such that one can interact with the output and also export it as a static image file (Fig. 3) (Ondov, Bergman & Phillippy, 2011).

Figure 3 Taxonomic classification of reads in an isolate using Centrifuge and Krona.

Read mapping and variant discovery

Mapping refined sequence reads to the reference genome is done by Snippy (Seemann, 2015b) which is a variant calling and core genome alignment pipeline. Firstly, BWA MEM (Li & Durbin, 2009) maps reads to the provided reference genome and they are manipulated with Samtools (Li et al., 2009). Variants are called from the resulting Binary Alignment Map (BAM) files using freebayes (Garrison & Marth, 2012) which takes short-read alignments in BAM formats and Phred+33 quality scores from each sample and determines the best combination of polymorphisms in each sample at each position in the reference genome with the aid of Bayesian variant detection model. The variants are annotated with SnpEff (Cingolani et al., 2012), with reports generated in variant call format (VCF), TAB, CSV, HTML, BED, GFF and FASTA files. A unified VCF file that contains core and full genome alignments of individual SNP output files are generated with Snippy-core. Furthermore, BAGEP leverages on a function in Snippy-core that allows the user to filter out problematic or repetitive regions core genomes of the samples such as Proline-Glutamate (PE) and Proline-Proline-Glutamate (PPE) gene families in M. tuberculosis by providing a BED file to omit these regions.

Antimicrobial resistant gene screening

BAGEP uses abricate (Seemann, 2015a) to screen the isolates for antimicrobial resistant (AMR) genes by using either one of the many AMR databases that comes with it. The great advantage about abricate is that these databases can be updated from the help option and the user can select which databases suits their analysis. The results from this step is stored in a tab-separated file which can be viewed in a variety of software.

SNP visualisation

One other key feature of BAGEP is the ability to generate an interactive SNP visualization that allows the end user to view substitution polymorphisms across core genomes in the population. In the pipeline, a custom R script makes use of vcfR (Knaus & Grünwald, 2017) and heatmaply (Galili et al., 2017), parses the VCF file output from Snippy-core to render a heatmap showing SNP positions in a HTML file (Fig. 4).

Figure 4 Interactive visualization of SNPs showing their positions across genomes.

(A) Twenty (20) M. tuberculosis genomes. (B) Twenty (20) Salmonella enterica serovar Typhi genomes.

The heatmap can be interacted with, with a number of tools such as zooming, hovering over the image to show SNPs and export a selected region as an image. This is very useful in a VCF file with high number of samples and reported SNPs as hovering around regions in the image can infer which position the polymorphism occur and in what sample.

Phylogeny

The final analysis done by BAGEP is the construction of a maximum-likelihood phylogenetic tree with IQ-TREE (Nguyen et al., 2015), this step uses the core genome alignment generated by Snippy-core with ultra-fast bootstrap value of 1000. The output tree is deposited in the ‘results’ directory and can be viewed and annotated with any tree viewing software such as Figtree or Interactive Tree of Life (ITOL).

Performance

Running BAGEP on 20 M. tuberculosis and 20 S. enterica serovar Typhi genomes with an 8 GB RAM, 2.5 GHz quad core laptop took 122 and 61 min to complete the analysis, respectively; M. tuberculosis has a 4.4 million base pair genome while S. enterica serovar Typhi has 4.81 million base pairs. BAGEP capitalizes on Snakemake’s multi-threading feature which implies that it can be deployed on laptops with greater performance or a computing server to improve its speed. Comparing BAGEP’s performance to Nullarbor pipeline on a benchmark computer using the same M. tuberculosis data sets, BAGEP took 122 min whilst Nullarbor spent 186 min to complete the same analysis. In addition to speed, BAGEP outperforms Nullarbor in detecting more accurate SNPs. To assess the robustness of BAGEP, we ran the pipeline on 57 M. tuberculosis genomes using a 16 GB RAM computer and it completed the run in 225 min (File S1).

Discussion

BAGEP was built around the core concepts of Snakemake which offers parallelization of several tasks in an orderly manner. It was designed to be user friendly, fast, customizable and reproducible. In addition, we wanted a pipeline that can be deployed on a personal computer and handle medium to large data and finally, an interactive tool for visualizing SNPs that were detected whilst running the pipeline. BAGEP requires minimal user input and runs from start to finish with a single command.

As the backbone of BAGEP is Snakemake, each rule is run in its own environment and allows the combination of other programming languages such as Python, R and Bash. This also implies that each rule can be customized to make it run faster depending on the configuration of the machine running it and basic understanding of tools used in the pipeline. For example, the rule that takes the longest time to run is the step where reads are mapped to a reference genome and variants are called. This is executed by Snippy and the number of threads can be increased to speed up the process.

This pipeline has been tested with M. tuberculosis and S. enterica serovar Typhi and it is suitable for genetically monomorphic or monoclonal pathogens such as Yersinia pestis, Bacillus anthracis, Mycobacterium leprae and Treponema pallidum due to the limited amount of variations in their core genome as compared to highly recombinant pathogens (Achtman, 2008).

For visualization, BAGEP generates a HTML file as output that contains observed SNPs in core genomes of the samples and the position where they can be found, in addition with dendrograms that highlight the relatedness of the genomes. The interactive image can be zoomed in and out, reveal SNP regions by hovering around the image with a mouse and also export the image as portable network graphics (PNG) image. This image can be easily interpreted by anyone with little or no knowledge in bioinformatics and gives a summary of the analysis.

It is worthy to note that BAGEP was compiled with speed in mind and ease-of-use and that is why the dependencies can be installed in a conda environment under the bioconda channel (Dale et al., 2018). Performance comparison to Nullarbor shows that BAGEP identified 459 SNPs in a particular isolate which yielded 598 SNPs with Nullarbor (it is important to note that BAGEP and Nullarbor uses the same tool, Snippy for variant detection). In other to validate the accuracy of the SNPs, we ran the same sample as raw reads (that is without undergoing QC steps with fastp or Trimmomatic) and the end result was 598 SNPs, this trend was seen in other samples as well. This indicates that the QC step in Nullarbor is not as effective as the one embedded in BAGEP, which will give rise to false positives when using Nullarbor. Furthermore, BAGEP is quicker than Nullarbor when analyzing the same set of samples. Therefore, standard population genomics of low recombinant or genetically monomorphic bacteria can be conducted with BAGEP as it handles the fundamental analysis needed.

Conclusions

We present BAGEP, a fully automated and scalable pipeline that is built on Snakemake framework. This pipeline will be useful for researchers in low-to-middle income countries and people with little or no bioinformatics skills in analyzing raw genomics data. In addition, this software is useful for people who have bioinformatics skills, but have little time performing laborious analysis on their data as BAGEP can provide a faster and automated way to analyze raw data in a user-friendly manner. This also gives them more time to focus on other activities without outsourcing data analysis.

It is effective for running medium to large data sets of paired-end raw reads of bacteria genomes, and has been tested with M. tuberculosis and S. enterica serovar Typhi. We have also shown that it is suitable for genetically monomorphic or monoclonal pathogens such as Yersinia pestis, Bacillus anthracis, Mycobacterium leprae and Treponema pallidum. Some of BAGEP’s advantages is that it is quick, identifies SNPs with better accuracy, easy to run and can be deployed on a mid-range laptop computer. Our future plan is to improve this pipeline by adding new features and roll out updates as we collaborate with other scientists.

Supplemental Information

Supplemental Information 1 Interactive heatmap visualization of SNPs and positions across the genome

Click here for additional data file.

We will like to appreciate the entire team at the African Center of Excellence of Genomics for Infectious Diseases (ACEGID), Redeemer’s University, Nigeria and the Lab of Viral Zoonotics (LVZ), Department of Veterinary Medicine, University of Cambridge, UK for establishing the partnership that birthed the idea behind BAGEP.

Additional Information and Declarations

Competing Interests

Author Contributions

Data Availability

Simon D.W. Frost is employed by Microsoft Research and is an Academic Editor for PeerJ.

Idowu B. Olawoye conceived and designed the experiments, performed the experiments, analyzed the data, prepared figures and/or tables, authored or reviewed drafts of the paper, and approved the final draft.

Simon D.W. Frost conceived and designed the experiments, authored or reviewed drafts of the paper, contributed to source code of the pipeline, and approved the final draft.

Christian T. Happi conceived and designed the experiments, authored or reviewed drafts of the paper, and approved the final draft.

The following information was supplied regarding data availability:

BAGEP source code and documentation is available at Github: https://github.com/idolawoye/BAGEP.

The Salmonalla enetrica serovar Typhi test data is available at Zenodo: Olawoye, Idowu Bolade, Frost, Simon, & Happi, Christian. (2020, March 27). The Bacteria Genome Pipeline (BAGEP): An automated, scalable workflow for bacteria genomes with Snakemake. (Version 1.0). Zenodo. http://doi.org/10.5281/zenodo.3731118.

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
