# Peer review of "The Bacteria Genome Pipeline (BAGEP): an automated, scalable workflow for bacteria genomes with Snakemake"

_PeerJ, doi:10.7717/peerj.10121_

## Round 0.1 · original submission · Major Revisions

As you can see, all reviewers agree on the need for you to expand your work as to compare it with other existing tools. Please address reviewers comments. Not just argue against them, try to adapt your work so as to satisfy their requests.

·

Basic reporting

The pipeline put forward by authors seems mostly suitable for the task at hand, which is assembly and visualiation of monomorphic bacterial WGS data. My main concern was that it did not take into account recombination for phylogenetics, which is very important for many bacteria. Authors state at the end of the discusison that the pipeline is only suitable for monomorphic bacteria: this should be clearly stated much earlier on, including in the abstract, so readers do not think this approach will work for bacteria that undergo any level of recombination.

Similarly, the abstract and the instriduction state "However, there is the need to analyze multiple genomes within a short time, in order to provide critical information about a pathogen of interest such as drug resistance, mutations and patient-to-patient transmission in an outbreak setting". I don't see how this pipeline addresses this need, as it does not do drug resistance prediction or transmission analyses, only SNP visualisation and phylgenetics. Thus, the need for another wrapper of existing software needs to be more clearly stated so the reader can better decide why to use this pipeline over a rival such as MTBseq or Nullarbor.

Experimental design

I was not able to install the pipeline on either an OSX laptop or a Linux sever. For both it hung at the 'solving environment' stage of the conda command. After 1 hour of wiating I canceled the install.
Both the laptop and the server have regularly used conda installs and others were checked afterwards to ensure it was the package and not the conda or hardware. This should be fixed so the software can be evaluated.

Authors state that the software has only been tested on a linux machine, yet say the target audience is LMIC researchers and those using a mid-range laptop. I would posit that most researchers that fall into this category are not using a Linux machine and thus the software should be tested on other platforms as well, or at least give instructions or pointers on their github to how to install virtual boxes or similar.

Line 123-125 states that dependancies come from bioconda but a bash script is needed for the R libraries. R libraries can be installed through conda so I dont see why this is 2 steps, especially if the authors are putting this forward as a '1 install/pipeline only' solution.

Similarly, I am unsure why the authors have a git install that then requires a conda environment for use, and ask that R already be installed. If using conda, they should use conda throughout the installation as a single command (e.g. conda install BAGEP), maybe from the bioconda channel, which would install the pipeline and the associated dependancies, including R and the R libraries. As it stands, it is not a single command install, which is what snakemake and conda were made to do.

Methods and github page do not indicate if parameters can be changed, such as model of evolution for IQ-TREE or cut-off parameters for snippy. This would be important as different bacteria require different parameters for phylogenetics. Without the ability to change IQ-TREE parameters, I see this as a massive drawback.

Validity of the findings

No comparison to existing software was undertaken. Authors state that MTBseq installationw as tried but failed. However, this is an M. tuberculosis-specific pipeline and not a general bacterial pipeline, which is what authors should be comparing to. The most obvious candidate for comparison would be nullarbor (https://github.com/tseemann/nullarbor) which undertakes most of what the authors are doing here, plush additional tasks and is created by the author of snippy. A comparison to this, to see the need for such a competing pipeline, should be done.

It has been shown previously that removal of contamination from read sets before mapping is required to reduce false SNPs for phylogenetics and other tasks (see Goig et al https://bmcbiol.biomedcentral.com/articles/10.1186/s12915-020-0748-z). This pipeline does not do this, and thus may gibve false results to the user. Authors should indicate why this is omitted or add a step that does this., However, contamination removal unfortunately tends to require large databases (this is why UVP is so large, as the authors state), so is incompatible with a laptop setup for such pipeline, which is one reason why such pipelines dont exist. Even still, authors need to justify why their pipeline doesnt do this, which likely will increase the rate of false SNP discovery.

Additional comments

I think the technological gap the authors are trying to fill here is admirable and necessary. Indeed many LMIC are being left behind in the WGS era because much analyses require high performance computing and standalone pipelines that are pathogen specific (see https://pubmed.ncbi.nlm.nih.gov/31209399/ for a discussion of this for M. tuberculosis, for example). I agree that a pipeline that bypasses this is required. However, there is always pitfalls in trying to create cross-pathogen, low resource pipelines, including omission of heavy processing steps (like contamination removal) that tend to still be very necessary, and not undertaking pathogen specific steps (like rtecombination removal or masking of repeat regions, which both have massive impacts on downstream analyses).
Thus, I think the idea for such a pipeline is comendable, it is tough to justify how this one addresses that issue and wont just give incorrect results due to the generalisability and low resource setup. Authors need to address these points to ensure incorrect results from their pipelines are not perpetuated in the resource community.

Reviewer 2 ·

Basic reporting

Introduction
Line 69. “as next generation sequencing gradually replaced Sanger sequencing”. NGS do not replaced Sanger sequencing. WGS is very useful for large scale works as epidemiology, infection control, contact tracing, surveillance etc, however most of research does not require WGS and Sanger sequencing is still an important tool for research. One of the problems of WGS are the ambiguous results due to sequences of low quality that can result from several incorrect procedures that frequently are solved/verified by Sanger sequencing. I suggest rephrase this statement as it can induce the readers in error.
Materials & Methods
In the MM should be included information about the genome sequences used to evaluate the pipeline. Their origin (published sequences or not – it is in the beginning of results but that information could be added here for contextualization), method used for DNA extraction (no need to be detailed), if the DNA was obtained from cultures or direct from clinical samples, were the sequences are deposited (database) and their accession numbers and study accession. Patient information should not be included.

Other remarks
Designation of the species: some inconsistency was observed. The species need to be mentioned in full at the first time in the main text (excluding abstract). The same for Salmonella. In the middle of the results appears MTB; should be maintained the terminology used from the beginning. If want to designate the species name abbreviated, please change at the very beginning.

MTBseq and the remaining pipelines/software’s – without italic please; there is no need. Italic should be reserved for the name of the bacterial species.

Experimental design

No comment.

Validity of the findings

The article meet PeerJ standards, with suggested improvements.

Annotated reviews are not available for download in order to protect the identity of reviewers who chose to remain anonymous.

·

Basic reporting

1. Authors should acknowledge, more recent work in the filed. For example when referring to GWAS (lines 92-93), Farhat et al, 2019 (Nature Communications).
2. The figures are in low resolution (for example the text in y axis is overriding itself in figure 2), and I would use better representation for SNPs along a chromosome (see example in Liu et al 2020, Scientific Advance).
3. Results does not demonstrate improvement in performance when compared to existing tools.

Experimental design

- When developing a new computational research methods / tools which aim to out-preform other existing methods, it is essential to compare performance in terms of accuracy, memory usage, running time, etc. This was not done here.
- Bacterial clinical samples tend to contain non-target contaminations, which generates noise and hampers downstream analyses. In order to address such contaminations, there are quality control measures that are used (example can be Centrifuge by Kim et al, 2016 Genome Research). Such measures are not included in the BAGEP pipeline.
- Details and instructions for installation of BAGEP are only referred to a git page, which should also be included in the MS.

Validity of the findings

- The data sets used for demonstration of the applicability of the developed tool is too small to make a good assessment (n=20).

Additional comments

The idea of making a 'light version' tool for WGS data of bacterial genomes is very compelling. There is a clear need for providing user friendly tools with minimum hardware requirements and high performances, and therefore I command the authors for giving it a try. I think the BAGEP tool has great potential and will help scientists in low income settings, however the MS lacks basic features for publishing a new computational pipeline.

---

## Round 0.2 · accepted · Accept

Before publication, however, I strongly recommend you incorporate into the manuscript the changes suggested by Reviewer 2. Once incorporated, please return your manuscript for publication. Please understand that your manuscript took longer than usually required to process due to the current pandemic situation. Thanks for publishing in PeerJ.

·

Basic reporting

No comment

Experimental design

No comment

Validity of the findings

No comment

Additional comments

All my comments have ben addressed sufficiently.

Reviewer 2 ·

Basic reporting

Minor corrections to the text:
L89. Please revise “single nucleotide polymorphisms (SNP)”
L90. Please revise “(indels) and/or translocations.”
L91. Please revise “ways to detect variants”
L92. Please revise “genomics of bacterial isolates”
L93. Please revise “bacterial species”
L94. Please revise “to understand evolution of bacterial genomes”
L98. Please revise “and/or”
L102. Please revise “bacterial genomes”
L110. Please revise “who are becoming users of WGS of pathogens”
L111. Please revise “also to provide”
L115. Please revise “income countries (LMICs)”
L180-181. Reference year are in bold. Please revise
L199. Please revise “The great advantage about Abricate”
L270-271. “This pipeline will be useful for researchers in low-to-middle income countries and people with little or no bioinformatics skills in analyzing raw genomics data.” I think that this software will be useful not only by the reasons pointed by the authors but also for those that have bioinformatics skills but no time available to perform such laborious analysis. As an example, prefer to outsource this kind of services to have the complete genome sequence due to lack of time. If this software can reduce the time spend for assembly and annotation making the process user-friendly will be an add on to this area leaving more time for other tasks. sentence could be improved in this sense.
L278-280. This sentence is repeated in line 270-271. Please revise.
L288. Please revise “Salmonella enterica serovar Typhy”

Experimental design

No comment

Validity of the findings

No comment

·

Basic reporting

No comment

Experimental design

no comment

Validity of the findings

no comment

Additional comments

Authors satisfied my concerns raised in the first round of review.